# Smart Responsive Microneedles for Controlled Drug Delivery

**DOI:** 10.3390/molecules28217411

**Published:** 2023-11-03

**Authors:** Zhenzhen Qi, Zheng Yan, Guohongfang Tan, Subhas C. Kundu, Shenzhou Lu

**Affiliations:** 1National Engineering Laboratory for Modern Silk, College of Textile and Clothing Engineering, Soochow University, Suzhou 215123, China; 20224015005@stu.suda.edu.cn (Z.Q.); 20225215103@stu.suda.edu.cn (Z.Y.); 20234015007@stu.suda.edu.cn (G.T.); 23Bs Research Group, I3Bs Research Institute on Biomaterials, Biodegrabilities, and Biomimetics, Headquarters of the European Institute of Excellence on Tissue Engineering and Regenerative Medicine, University of Minho, AvePark, Guimaraes, 4805-017 Barco, Portugal; kundu@i3bs.uminho.pt

**Keywords:** microneedles, transdermal delivery, smart responsiveness, controlled drug release

## Abstract

As an emerging technology, microneedles offer advantages such as painless administration, good biocompatibility, and ease of self-administration, so as to effectively treat various diseases, such as diabetes, wound repair, tumor treatment and so on. How to regulate the release behavior of loaded drugs in polymer microneedles is the core element of transdermal drug delivery. As an emerging on-demand drug-delivery technology, intelligent responsive microneedles can achieve local accurate release of drugs according to external stimuli or internal physiological environment changes. This review focuses on the research efforts in smart responsive polymer microneedles at home and abroad in recent years. It summarizes the response mechanisms based on various stimuli and their respective application scenarios. Utilizing innovative, responsive microneedle systems offers a convenient and precise targeted drug delivery method, holding significant research implications in transdermal drug administration. Safety and efficacy will remain the key areas of continuous efforts for research scholars in the future.

## 1. Introduction

Traditional administration routes, such as subcutaneous injection and oral intake, are widely employed to deliver most pharmaceuticals [1]. Compared to conventional methods, transdermal drug delivery mitigates needle phobia and pain, enhancing patient compliance [2]. Furthermore, transdermal delivery typically requires lower drug doses than oral intake, and it can circumvent the adverse effects caused by the unstable absorption and metabolism of drugs in the gastrointestinal tract [3].

In recent years, there has been significant attention on minimally invasive transdermal drug delivery platforms based on microneedles [4]. Microneedles essentially consist of a patch with numerous micrometer-length needles, with needle lengths exceeding the thickness of the skin’s stratum corneum. As depicted in Figure 1, once inserted into the skin, microneedles can easily penetrate the stratum corneum, facilitating the direct delivery of drugs to the epidermis and dermis, subsequently being absorbed into the circulation through microvessels and lymphatic vessels [5].

In 1976, Gerstel introduced the concept of microscale arrays for instant penetration of the stratum corneum in a patent. However, at that time, due to a lack of manufacturing technology, it was impossible to manufacture and commercialize microneedles. The emergence of microelectronics tools used in the semiconductor industry in the 1990s renewed the interest of scientists in manufacturing microneedles [2]. Since then, various materials have been used for microneedle fabrication, such as ceramics [6], stainless steel [7], polymers [8], silicon [9], silk [10,11,12], and sugars [13], among others. The earliest preparation method of microneedles is MEMS technology, and on this basis, laser cutting, photolithography, wet and dry etching, micro-molding, 3D printing, and so on have been developed [14,15]. One of the most common, simplest, and most convenient methods is the micro-molding method. Figure 2 illustrates the developmental progress of microneedles over different periods. Initially designed for delivering low-dose drugs, microneedles are now employed in a wide range of therapeutic applications, including the delivery of hydrophilic high-dose medicines [16], nanoparticles and microspheres [17], peptides and protein drugs [18], various types of vaccines [19], and DNA [20], among others.

Conventional soluble microneedles can only release drugs directly into the skin according to predetermined design protocols, which may not meet complex physiological requirements. Therefore, smart microneedles with environmental responsiveness must be developed using non-soluble microneedle substrates. In recent years, responsive strategies have been applied to drug delivery. Intelligent drug delivery refers to the drug-carrying smart response materials based on changes in physiological stimuli (pH value, temperature, enzymes, different biomolecules, etc.) or external stimuli (electric field, current, magnetic field, light, mechanical external forces, etc.) to deliver drugs to the biological system. Environmentally responsive microneedles can release their loaded drugs under specific stimuli, creating controlled release patterns, thus making them safer and more efficient than traditional microneedles. The number of publications and patents on responsive microneedles has increased significantly in recent years, and we obtained the following data by conducting keyword searches on Pubmed and CPC systems (Figure 3). The number of patent applications has been relatively low in recent years, perhaps because the latest data have not been updated and patents take longer to review.

Smart microneedles promise to break the bottlenecks of inefficient delivery, inevitable drug waste, and simple delivery modes and are considered to play an essential role in a wide range of biological fields. Drug delivery systems based on glucose response have attracted widespread attention, and Gu’s team [21,22,23,24] developed a transdermal polymer microneedle patch for insulin delivery to achieve glucose regulation. In addition to blood glucose control, intelligent responsive microneedles have been studied in wound healing [25,26,27] and cancer treatment [28,29,30].

This review has compiled a comprehensive overview of intelligent responsive microneedles used for drug delivery, which respond to various external and internal stimuli. We have provided detailed discussions on their substrates, fabrication methods, action mechanisms, and application scenarios. The current development status and future development trends of smart microneedles are also discussed.

## 2. Endogenous Stimulus-Responsive Microneedles

### 2.1. Response to Glucose Concentration

In 2021, the International Diabetes Federation released the Global Diabetes Atlas, which reported that 537 million adults aged 20 to 79 are living with diabetes, accounting for 10.5% of the global population in this age group. Glucose-responsive microneedles, characterized by their painless, minimally invasive, and glucose-sensitive properties, are showing promise as potential replacements for injection therapies.

Glucose-responsive materials can specifically bind to glucose, making them suitable for use as glucose-sensitive models [31]. Three typical glucose-sensitive models include phenylboronic-acid-based polymers (PBA) [32], glucose oxidase (GOx) [33], and concanavalin A (Con A) [34]. The following sections provide detailed introductions to each of these glucose-sensitive models.

#### 2.1.1. Glucose-Sensitive Microneedles Based on PBA

The mechanism of action of PBA with glucose was first reported by Kuivila and colleagues in 1954. It exploits the characteristic binding of boronic acid with polyhydroxy compounds. Boronic acid is attached to the benzene ring to enhance responsiveness, forming phenylboronic acid that participates in the reaction. The binding of PBA with glucose in an aqueous solution is depicted in Figure 4. The boron atom forms a tetrahedral structure with polyhydroxy compounds, converting phenylboronic acid from an uncharged to a negatively charged state. Hydrolyzed PBA in this state readily forms boronate ester bonds with glucose molecules. Simultaneously, the presence of polyhydroxy structures in glucose significantly improves the water solubility of PBA. Based on PBA’s charge properties, hydrophilicity changes, and the specific binding between the phenylboronic and hydroxyl groups, glucose-sensitive materials can be designed using PBA as a foundation.

The formation of glucose-PBA complexes increases negative charge, reducing the electrostatic interactions between negatively charged insulin and the microneedle substrate. This enhances the homo-repulsive forces between microneedle substrate molecules, increasing the swelling of the microneedles and thereby promoting insulin release. Based on this mechanism, Gu and colleagues [36] fabricated PBA microneedles loaded with insulin through in-situ photopolymerization (Figure 5a). When PBA interacts with glucose, the microneedles undergo swelling, and as the swelling increases, it accelerates the release of insulin captured within the polymer network.

Microneedles prepared through in-situ polymerization techniques result in the formation of non-degradable chemically crosslinked polymer networks during fabrication. This process introduces the risk of physiological exposure to toxic unreacted monomers, crosslinkers, or polymer initiators. Using glucose as a dynamic exchange substance to replace binding sites formed by diol-structured polymers such as boronic acid phenyl esters can also establish glucose-responsive drug delivery models based on dynamic gel networks. Zhou and colleagues [37] combined PBA-modified synthetic polymers with tetra-armed polyethylene glycol macromonomers partially end-capped with diols, generating dynamic hydrogels with glucose-responsive properties (Figure 5b). These dynamic covalent hydrogels were loaded into microneedle molds by centrifugation, leading to the fabrication of microneedle patches. In the presence of glucose, the insulin loaded within the microneedles was released at an accelerated rate. The practicality of insulin encapsulation and in vivo delivery was validated in diabetic rats.

The pKa of PBA is 8.8–8.9 [38], which makes it prone to aggregation in the body, thereby limiting its ability to bind to glucose molecules to some extent [39]. To reduce the usage of glucose-responsive materials, microneedle patches with variable crosslinking densities in a core-shell structure [40], composite semi-interpenetrating hydrogel microneedles [41], and double-layered porous microneedle patches [42] have emerged. Additionally, modification of the structure of PBA can be employed to enhance its glucose sensitivity [43]. Furthermore, there is also concern about drug inactivation during the preparation and loading processes of phenylboronic-acid-based smart microneedles, necessitating the selection of relatively gentle preparation and storage conditions [35].

#### 2.1.2. Glucose-Sensitive Microneedles Based on GOx

Glucose oxidase can convert β-D-glucose into D-glucose acid and hydrogen peroxide, creating a locally acidic environment. Responsive models can be designed using electrostatic and host–guest interactions based on the acidic environment generated through catalysis. The initial glucose-responsive materials were developed by combining glucose oxidase as the sensing component with pH-sensitive hydrogels as the scaffold. The immobilization of GOx imparts glucose-responsive properties to pH-sensitive materials. Lu’s team [44,45] achieved cationic silk fibroin by chemically modifying it with ε-poly-(L-lysine). This pH-sensitive material is prepared through enzyme crosslinking with horseradish peroxidase and hydrogen peroxide. The swelling degree of this material increases with the decrease in pH value and increases with the increase in glucose concentration.

After the enzyme oxidizes glucose, hydrogen peroxide can be utilized to design glucose-sensitive models through redox mechanisms. As shown in Figure 6, Xu et al. [46] proposed the modification of nanoparticles with 4-(imidazol-1-ylmethyl)phenylboronic acid pinacol ester (ICBE). GOx and insulin were further encapsulated within the nanoparticles, and they achieved insulin retention within the nanoparticles through host–guest complexation with α-cyclodextrin. In the presence of hydrogen peroxide, the phenylboronate on the surface of the nanoparticles undergoes oxidation, disrupting the host–guest complexation, leading to the decomposition of nanoparticles, and subsequently releasing the pre-loaded insulin.

Yu et al. [47] developed an insulin-delivery device containing glucose-responsive vesicles. These vesicles were composed of hyaluronic acid combined with the hypoxia-sensitive 2-nitroimidazole (NI) and loaded with insulin and GOx. NI is a hydrophobic component that can be converted into hydrophilic 2-aminomethyl imidazole under hypoxic conditions. In a hyperglycemic state, the localized hypoxic microenvironment caused by glucose oxidase activity promotes the reduction of NI, triggering rapid vesicle dissociation and subsequent insulin release.

Sensitive models based on GOx need to consider enzyme inactivation caused by changes in local oxygen concentration and pH during the reaction. Lu et al. [48] confirmed that when silk membranes loaded with GOx were stored at 37 °C for 10 months, the activity of GOx was hardly reduced. Changes in the in vivo microenvironment may impact the activity state of surrounding cells and tissues [49], disturb the internal balance, and lead to disruptions in surrounding tissues [50]. These safety risks may limit glucose oxidase’s applicability as a sensitive model.

#### 2.1.3. Glucose-Sensitive Materials Based on ConA

Con A is a lectin protein found in canavalia that exhibits a reversible strong affinity for non-reducing α-D-mannose, α-D-glucose, N-acetyl-D-glucosamine, and polysaccharides. This property makes it suitable for creating glucose-sensitive systems [51]. In Con-A-based systems, the mechanism for glucose-sensitive drug release is as follows: Con A is either embedded or immobilized in the glucose-containing microneedle substrate. Insulin or other model drugs are loaded into the microneedle matrix during or after the formation of the three-dimensional structure. When the microneedle substrate comes into contact with glucose, Con A rapidly competitively binds to glucose, displacing the binding sugar moieties on the material and disrupting the original 3D structure and drug release. With increasing glucose concentration, more glucose binds to Con A, releasing more drugs, making Con A-modified materials potentially suitable for use as glucose-sensitive drug delivery systems.

Yin et al. [52] designed a chitosan derivative as a polymer ligand for Con A, exhibiting a strong affinity for Con A, and utilized genipin crosslinking to prevent Con A leakage. Glucose-responsive hydrogels were prepared using a reverse emulsion crosslinking method. In vitro experiments demonstrated that insulin release could reversibly respond to different glucose concentrations, meeting the requirements for both rapid and sustained release. To address the poor water solubility and stability of Con A, Kim et al. [53] employed mono-methoxy polyethylene glycol to pegylate Con A with nitrophenyl carbonate. The results showed improved solubility and stability of Con A after modification, with the maximum binding affinity for glucose achieved when pegylation was at 50%. However, compared to hydrogels, microgels, and microparticles, glucose-sensitive microneedles based on Con A have been scarcely mentioned.

Glucose-sensitive models based on Con A still have limitations, such as immunogenicity, low solubility, poor stability, and relatively long response times [54,55], and further exploration and research are needed in this regard.

### 2.2. Response to pH

Extensive research has been conducted on pH-responsive systems among various stimuli. The normal pH of the skin ranges from 4.5 to 6.0, depending on several intrinsic factors (such as age, location, individual differences, etc.) and extrinsic factors (such as skin cleanliness, etc.) [56]. There are primarily two strategies for developing pH-responsive drug delivery systems.

The first strategy involves structural or solubility changes in polymers containing ionizable functional groups [57]. Various ionizable groups in polymers (such as carboxylic acid and amino groups) can undergo protonation at low pH, disrupting the hydrophilic–hydrophobic balance and triggering a rapid change in polymer structure or solubility, thereby achieving pH-responsive drug release. Glutamic and aspartic acid side chains in silk protein contain –COOH groups, which lose H^+^ in a neutral solution and form –COO–, resulting in a negative charge. Tan et al. [45] used grafting poly(ε-poly-L-lysine), a cationic peptide composed of 25–30 lysine residues, to obtain acid-sensitive cationic silk molecules. The main link branches of the modified fibroin protein have side chains of –NH_2_ group, and the whole side chain has no charge. In the presence of H^+^, –NH_2_ can be converted to –NH_3_^+^. In solutions with higher acidity (i.e., lower pH values), the increase in H^+^ expands the interchain spacing due to electrostatic repulsion, resulting in higher swelling. The expansion reaction weakens when the pH increases (becomes more alkaline). This pH-responsive expansion occurs only at pH < 7. Li et al. [58] coated a pH-responsive polyelectrolyte multilayer film (PEM) on the surface of polycaprolactone (PCL) microneedles using a layer-by-layer assembly approach. Poly(L-lysine) modified with dimethyl maleic anhydride (PLL-DMA), a charge-reversible polymer responsive to acid, was incorporated into the PEM. Initially, a polydopamine layer was coated on the PCL microneedle surface, and after protonation treatment, it formed a negatively charged transition layer (PLL-DMA/polyethyleneimine). Finally, a positively charged gene-loaded layer (p53 expression plasmid/polyethyleneimine) was reassembled to form the PEM. As shown in Figure 7a, in an acidic skin environment, the negatively charged PLL-DMA can transform into positively charged PLL on the microneedle surface, leading to surface collapse and promoting the release of the outermost gene-loaded layer containing the p53 gene.

Another strategy relies on the cleavage or degradation of acid-labile bonds. Chemical bonds such as hydrazones, esters, pyridines, imines, and ketals are stable at neutral pH but cleave under acidic conditions [59]. Maaden et al. [60] modified the surface of silicon microneedles with amino groups to achieve pH-sensitive microneedles, which were subsequently derivatized into pyridine groups. Ovalbumin was coated onto the pH-sensitive microneedle array. Ovalbumin was coated within the pH range between the surface pKa and the pI of protein, with the protein being negatively charged and the microneedle device surface being positively charged. Upon insertion into the skin at a pH higher than the surface pKa, the coated microneedles could rapidly release the protein while maintaining its antigenicity. Some inorganic small molecules, such as NaHCO_3_ and CaCO_3_, can also be used as pH-responsive materials. Under acidic conditions, H^+^ can react with substances like NaHCO_3_ and CaCO_3_, generating CO_2_ gas, which creates pores in the drug carrier, facilitating drug release. Hu’s team [61] recently designed polyethylene glycol-co-poly(lactic acid-co-glycolic acid) (PLGA) that was hybridized with calcium carbonate to form nanoparticles (6s-NPs (CaCO_3_)) and loaded onto microneedles, rendering the microneedles acid-responsive. As shown in Figure 7b, these nanoparticles only release under acidic conditions caused by arthritis, enhancing drug delivery specificity. The responsiveness of these substances is often irreversible.

**Figure 7 molecules-28-07411-f007:**
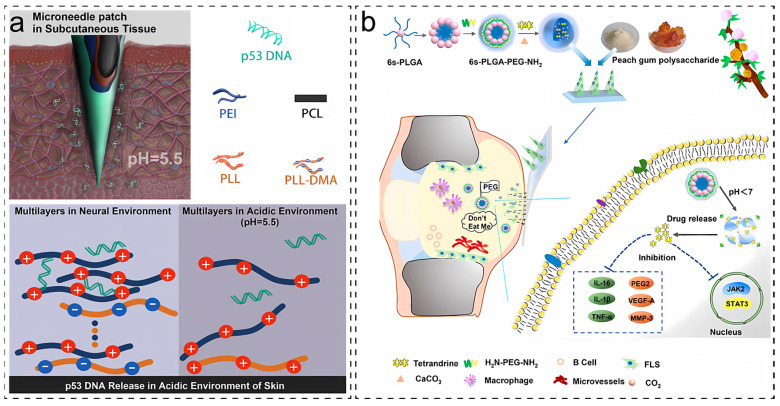
Preparation of pH-responsive microneedles. (**a**) Schematic diagram of self-assembled pH-responsive microneedles. (**b**) MN loaded in response to the pH of the nanoparticles; under acidic conditions, the nanoparticles are destroyed, and CO_2_ is generated. Image is used with permission of [58,61].

### 2.3. Response to Reactive Oxygen Species (ROS)

Reactive oxygen species (ROS) are essential signaling molecules in biological metabolism and intercellular communication, primarily including hydrogen peroxide (H_2_O_2_), singlet oxygen (^1^O_2_), superoxide (•O_2_^−^), and hydroxyl radicals (•OH). In various pathological processes such as diabetes, inflammation, and cancer, ROS levels are significantly elevated due to metabolic abnormalities. By leveraging the differences in ROS levels between healthy and diseased tissues, various ROS-responsive groups have been developed for ROS-sensitive drug delivery systems [62,63]. There are two main response mechanisms commonly used for ROS-responsive polymers.

One is the change in hydrophilicity and hydrophobicity under ROS action. ROS can oxidize sulfur elements and increase their valence. Oxygen and sulfur atoms form covalent bonds, and polar groups interact with water molecules through hydrogen bonds, thereby triggering the hydrophobic–hydrophilic transition of the polymer main chain. Wang et al. [64] developed a main-chain degradable amphiphilic polymer based on bis(6-hydroxyhexyl) 3,3′-selenodipropionate with ROS-responsive β-selenocarbonyl groups in the hydrophobic segment of the main chain. At a 1 mM H_2_O_2_ level, bis(6-hydroxyhexyl) 3,3′-selenodipropionate was oxidized to bis(6-hydroxyhexyl) 3,3′-selenodipropionate salt, which was then eliminated to form 6-hydroxyhexyl acrylate, and ultimately, 90% of the polymer was eliminated after 24 h. This polymer may be used in the future to prepare responsive microneedles.

Another approach is for polymers to undergo structural changes in response to ROS. ROS can react with chemical structures such as sulfur ketones, phenylboronic acid/ester, ethylene disulfides, and pyroglutamic oligomers, leading to the breakage of these structures [65]. Based on this principle, Zhang et al. [66] prepared ROS-responsive microneedles using polyvinyl alcohol (PVA) crosslinked with a connector containing bisphenylboronic acid. Clindamycin was encapsulated within the PVA. The above materials were incubated in a phosphate buffer solution (pH 7.4) containing 1 mM H_2_O_2_ at 37 °C. Due to the oxidation of the crosslinker, rapid material degradation and sustained release of clindamycin were observed. Microneedle substrates displayed faster degradation and drug release rates in 10 M H_2_O_2_, further confirming the ROS-dependent degradation of the microneedle substrate. Bi et al. [67] developed a detachable and H_2_O_2_-responsive hydrogel microneedle. The hydrogel was formed through dynamic covalent phenylboronate ester linkage between tannic acid esters and phenylboronic-acid-modified hyaluronic acid. The drug release rate of MTX/EGCG microneedles was highest at 1.0 mM H_2_O_2_, reaching 100% within 48 h.

### 2.4. Response to Enzymes

Enzymes are a class of biological macromolecules found within living organisms that possess catalytic activity and specific spatial conformations. Due to their temperature sensitivity, enzymes typically catalyze reactions at lower temperatures within an aqueous environment. Enzymes have been applied in stimulus-responsive polymer microneedle systems primarily through enzyme-responsive polymer materials or enzyme-reactive biomaterials acting as targeting substrates. Changes in enzyme expression levels trigger substrate chemical structure alterations, thereby achieving drug delivery effects [68].

Enzymes are categorized into seven types: oxidoreductases, transferases, hydrolases, lyases, isomerases, synthases, and ligases [69]. Among them, hydrolases and oxidoreductases have received more attention in applications. Thrombin, a serine protease similar to trypsin, plays a crucial role in the coagulation system and can convert soluble fibrinogen into insoluble fibrin. Thrombin-responsive systems based on thrombin-cleavable peptides have garnered significant interest due to their high sensitivity and rapid response rates. As shown in Figure 8a, in a system developed by Zhang et al. [70], heparin (HP) was conjugated to the main chain of hyaluronic acid (HA) through UV-initiated polymerization, and a thrombin-cleavable peptide was introduced as a linker. First, the cleavable peptide formed an amide bond with methacrylated HA (m-HA), and then HP was covalently bound to the cysteine residues of the peptide. When thrombin levels rise, the thrombin-cleavable peptide is cleaved, triggering the release of HP in a thrombin-responsive manner. The released HP can inhibit coagulation by inactivating thrombin, minimizing the risk of spontaneous bleeding. As shown in Figure 8b, Ye et al. [71] proposed a “drug A in carriers formed by incorporation of drug B” strategy to enhance antitumor immune responses. 1-Methyl-DL-tryptophan (1-MT, an indoleamine 2,3-dioxygenase (IDO, an immunosuppressive enzyme) inhibitor) was covalently conjugated with hyaluronic acid to form an amphiphilic structure (m-HA), which self-assembled into nanoparticles and encapsulated antibodies (aPD1, anti-programmed cell death protein 1 (PD1, an immunosuppressive receptor on tumor lymphocytes)). Subsequently, these nanoparticles were integrated into the microneedle system. In the tumor microenvironment, overexpressed hyaluronidase (HAase) digested the HA within the microneedles, promoting drug release. The synergistic delivery of aPD1 and 1-MT induced effective and sustained antitumor effects enhanced T-cell immunity, and reduced immune suppression in the tumor environment.

The advantages of enzyme-responsive microneedle systems lie in their ability to leverage differences in enzyme expression levels between pathological microenvironments and typical physiological environments to achieve targeted therapy while minimizing the potential toxicity of drugs to normal cells and tissues. However, relying solely on enzyme reactions can make it challenging to achieve complete drug control and may involve the risk of premature drug release. Therefore, most enzyme-responsive drug delivery systems require synergistic use with other responsive materials. Yu et al. [72] developed a soluble microneedle system based on α-amylase and loaded with levofloxacin-containing dopamine nanoparticles, combining enzyme degradation, antibiotic action, and photothermal treatment. The α-amylase in the microneedles disrupts extracellular polymer substances, while the dopamine nanoparticles loaded with levofloxacin achieve antibiotic and mild photothermal therapy under 808 nm laser irradiation.

As shown in Table 1, in addition to the work specifically described above, we have outlined several responsive microneedles triggered by internal stimuli, outlined their responsive materials, loaded drugs and application scenarios.

## 3. Exogenous Stimulus-Responsive Microneedles

### 3.1. Response to Temperature

Temperature response is a crucial aspect of intelligent biomaterials, allowing for transdermal drug delivery via microneedles by controlling stimulus temperature, application site, and duration, and utilizing heat-responsive materials. Temperature-responsive materials play a pivotal role in temperature-responsive microneedles. These materials utilize the property of thermosensitive polymers to undergo phase transitions based on temperature changes, enabling controlled drug release. Currently, researched temperature-responsive materials include poly(N,N-dimethylaminoethyl methacrylate) (PDMAEMA), poly(N-isopropylacrylamide) (PNIPAM), and poly(methyl methacrylate) (PMMA) [89]. These polymers comprise hydrophobic and hydrophilic segments, forming amphiphilic block copolymers. By controlling the chain lengths of these copolymers, their morphological transformation can be effectively regulated.

Human skin temperature typically remains within the range of 35–37 °C, and in certain pathological conditions, skin temperature may vary due to the type and progression of diseases [90]. Therefore, changes in skin surface temperature can serve as effective stimuli for temperature-responsive microneedles.

In temperature-responsive polymer microneedle systems, changes in the polymer’s hydrophilicity and hydrophobicity lead to segment folding and aggregation as the temperature rises. Thermosensitive polymers with lower critical solution temperatures, such as cellulose, xyloglucan, chitosan, collagen-mimetic peptides, elastin-like protein polymers, and PNIPAAm, are all characterized by biocompatibility and degradability. They can undergo a sol-gel transition above their critical solution temperature and are widely used for temperature-sensitive drug release applications. PNIPAAm, which contains hydrophilic amide and hydrophobic isopropyl groups along its polymer chains, undergoes a phase transition in its aqueous solution at approximately 33 °C, transitioning from a homogeneous to a phase-separated system.

Chemically crosslinked PNIPAAm hydrogels experience a sudden volume contraction at around 32 °C upon heating. Based on this principle, Chi et al. [91] developed a multifunctional microneedle patch for promoting wound healing. Vascular endothelial growth factor (VEGF) was encapsulated within temperature-responsive PNIPAAm hydrogel microneedles. Inflammation-induced local skin temperature elevation induced the contraction of the PNIPAAm hydrogel microneedles, facilitating the controlled release of VEGF at the wound site. Leveraging the advantage of PNIPAAm, which can reversibly switch between a linear state and a coiled state with temperature variation, Li et al. [92] created a physically entangled crosslinked hydrogel microneedle patch using gelatin and carboxyl-terminated PNIPAAm as matrix materials. Gelatin-PNIPAAm exhibits a sol-gel transition property triggered by temperature changes and interacts appropriately with drug molecules, slowing down the crystallization rate of insulin in solution. With regular body temperature above the low critical phase transition temperature of gelatin-PNIPAm (31.3 °C), the system rapidly transitions from a gel to a sol state, effectively delivering drug microspheres into the skin within seconds and controlling drug release within the skin.

Furthermore, a few heat-responsive microneedles achieve drug delivery by melting low-critical-temperature polymers through photothermal catalysts. Common photothermal catalysts include metal sulfides, indocyanine green, black phosphorus, and polymer nanoparticles. Polycaprolactone (PCL) is an excellent temperature-responsive component with a melting point of 60–63 °C. Exploiting this characteristic, Hao et al. [93] designed a soluble microneedle system for the treatment of skin cancer. Initially, they performed ring-opening polymerization of ε-caprolactone with monomethoxy polyethylene glycol to prepare a diblock copolymer, monomethoxy-poly(ethylene glycol)-polycaprolactone (MPEG-PCL). Subsequently, they fabricated MPEG-PCL nanoparticles loaded with indocyanine green and 5-fluorouracil using a double-emulsion approach. These nanoparticles were then loaded into hyaluronic acid microneedles. After insertion into the skin, the nanoparticles were delivered into the body due to the dissolution of the hyaluronic acid. Following 5 min of irradiation with 1.5 W/cm^2^ near-infrared light (808 nm), the temperature at the tumor site increased to 65 °C, leading to the dissolution of MPEG-PCL nanoparticles and subsequent release of 5-fluorouracil.

### 3.2. Response to Light

Light offers a versatile source of adjustable intensity energy through modulating its wavelength, power intensity, exposure time, and beam diameter. Moreover, it allows for precise control and operation within specific areas, making it widely applicable and extensively researched in biomedicine. Light-responsive biomaterials can be categorized into four main types: photoisomerization materials, photothermal materials, photolysis materials, and photopolymerization materials. Photoisomerization materials undergo reversible conformational changes upon exposure to ultraviolet and visible light, without involving alterations in chemical bonds. Photothermal materials can absorb photon energy and generate heat through vibration or energy-level transitions, making them valuable for controlling drug delivery. When exposed to light stimulation, photolysis materials have their photosensitive chemical bonds within the drug delivery platform broken, allowing for the release of active components. Photopolymerization materials undergo crosslinking when exposed to light and have been applied in the preparation of in-situ crosslinked polymers and in the field of tissue engineering.

Hardy et al. [94] achieved photoresponsive ibuprofen conjugates by conducting esterification between the carboxylic acid on ibuprofen and 1,5-dimethoxyanthracene. Microneedles loaded with these conjugates released ibuprofen in a controlled manner for an extended period (over 160 h) upon exposure to ultraviolet light at 365 nm. However, the application of ultraviolet light is limited in photoresponsive microneedle systems due to its shorter wavelength, intense tissue penetration, and potential tissue damage. The most common light stimulus in photoresponsive polymer microneedle systems is near-infrared light (NIR) due to its deeper penetration and minimal tissue photodamage.

The NIR responsiveness of polymers is endowed by photosensitive protective groups capable of absorbing photons and converting them into thermal energy, leading to the breaking of molecular bonds or carrier melting. Controlled irradiation with NIR light enables the controlled degradation of polymer structures, thereby achieving the controlled release of a specific dose of active substances.

IR780 iodide, an amphiphilic cationic compound with maximum absorbance in the spectral range of 750–830 nm, and maintains excellent photostability due to the rigid cyclohexene ring in the center of its heptamethine chain and chlorine atoms. As a widely used and highly safe fluorescent dye for cell imaging, IR780 is extensively applied in photothermal therapy. Su et al. [95] loaded IR780 iodide, serving as a photothermal converter, and molecularly engineered peptide W379, acting as an antimicrobial agent, into soluble polyvinyl pyrrolidone microneedle patches, subsequently coated with the phase-change material 1-tetradecanol. When exposed to near-infrared light, IR780 induces photothermal conversion, causing the melting of 1-tetradecanol (melting point: 38 °C), leading to the dissolution of the microneedle patch and consequently releasing loaded W379 peptide into the surrounding area.

Due to their excellent photothermal properties, biocompatibility, biodegradability, and low toxicity, Fe_3_O_4_ nanoparticles have received widespread attention in the biomedical field, particularly for targeted drug delivery. Under light exposure, Fe_3_O_4_ nanoparticles rapidly increase in temperature, enabling enhanced drug release rates for thermosensitive microneedles. Cui et al. [96] combined photothermal responsiveness with temperature sensitivity for drug control. In this system, Fe_3_O_4_ nanoparticles acted as photothermal catalysts, while gelatin was the microneedle substrate, imparting its temperature sensitivity. Upon near-infrared irradiation, the temperature of this microneedle patch rapidly increased by 40 °C within 1 min, resulting in a 70% release rate of the model drug (doxycycline hydrochloride) from the microneedles after 20 min of irradiation.

By applying NIR-responsive microneedles, apart from using photothermal converters to transform light energy into heat for matrix melting, non-reactive substances can also be utilized through photolysis. Effective control of covalent-bond breaking between molecules is achieved by adjusting the total applied energy, leading to the release of active drugs. Graphene oxide (GO) exhibits excellent absorbance in the NIR region, making it highly effective for photothermal applications. It serves as a carrier for drug delivery and can also act as a filler to enhance shape memory functionality in materials. Therefore, Fan et al. [97] introduced GO into PVA microneedle patches, imparting them with shape memory capabilities and controlled drug release under NIR irradiation (beam diameter: 0.5 cm, 1.5 W/cm^2^). In response to NIR irradiation, the oxygen-containing groups on GO undergo a reduction reaction, breaking the hydrogen bonds between GO and PVA, causing the microneedles to expand and release the drug fully.

### 3.3. Response to Electricity

Electrical stimulation is a signal factor that is controllable, monitorable, and can be applied remotely. The development of wearable energy-harvesting monitoring devices has been rapid. Loading drugs into electrically responsive carriers enables effective integration between microneedle drug delivery systems and electronic, informational, and smart technologies. There are four main mechanisms and approaches for electrically responsive drug delivery: current-activated drug carriers, electrochemical drug carriers, electroporation, and iontophoresis. Considering factors such as safety and drug delivery effectiveness, particularly for large molecules and vaccines, directly activating drug carriers with electric current appears to be the most feasible approach. Typically, conductive polymers are used as drug carriers, including materials such as conductive hydrogels, polypyrrole, polyaniline, and graphene, among others [98].

To confer electrical responsiveness to silk fibroin, Qi et al. [18] performed thiolation modification on silk fibroin to prepare electrically responsive silk protein microneedles. They achieved reversible electrical-responsive insulin release by utilizing the redox reaction between thiol groups and disulfide bonds under the influence of an electric current. Graphene oxide was incorporated into the microneedle carrier to enhance the electrical responsiveness and drug release rate.

As an intelligent, electrically responsive material, polypyrrole can load and release drugs through the conversion between its oxidized and reduced states, enabling the encapsulation and release of anions, cations, and neutral drugs. In the study by Yang et al. [99], a dual-electrode microneedle patch (t-EMNP) based on polylactic-acid–platinum–polypyrrole (PLA–Pt–PPy) was used for the treatment of specific dermatitis. The t-EMNP comprised two microneedle arrays: a drug-loaded PLA–Pt–PPy array and a PLA–Pt array. The PLA–Pt array was prepared by spray-coating Pt onto the surface of polylactic acid microneedles, followed by the deposition of drug-loaded PPy using oxidative deposition. A drug-doped PPy film was then deposited on the PLA–Pt array to create the drug-loaded PLA–Pt–PPy array. The PLA–Pt array and PLA–Pt–PPy array were connected to a power source’s positive and negative terminals, forming a completely electrically controlled transdermal drug delivery system. During the drug release process, the PLA–Pt–PPy MN array acted as the cathode in the reduction reaction, allowing anionic drugs to be released from the PPy into the skin under a certain voltage.

Those mentioned above electrical responsive microneedle patches all require an external power source. Recently, researchers have integrated nanogenerators, including triboelectric nanogenerators (TENG) and piezoelectric nanogenerators (PENG), with electrically responsive microneedles. Nanogenerators offer several advantages compared to traditional batteries: they are lightweight and flexible, allowing for easy integration with drug delivery devices and attachment to the skin. Additionally, nanogenerators collect biomechanical energy and convert it into electrical energy, eliminating the issue of energy depletion.

Yang et al. [100] designed and fabricated an integrated self-powered and controllable drug delivery system based on a piezoelectric nanogenerator and microneedle patch. This patch consists of two microneedle arrays: one with a polylactic-acid–gold–polypyrrole (PLA–Au–PPy) array loaded with dexamethasone as the working electrode (WE), and another with a polylactic-acid–gold (PLA–Au) array as the counter electrode (CE). The piezoelectric nanogenerator collects biomechanical energy through joint bending or tapping to generate electrical signals. The microneedle patch, owing to the reversible electroresponsive oxidation–reduction properties of the PPy film, can respond to electrical stimulation to release model drugs. TENG can also be integrated with the microneedle patch to convert mechanical energy from finger sliding into electrical power and deliver electrical stimulation through microneedles [101].

### 3.4. Response to Magnetic

The magnetic wave stimulation system is a non-biotoxic signal factor that can be remotely adjusted and mainly generated by alternating or fixed magnetic fields. In magnetic-responsive drug release systems, magnetic nanoparticles are commonly used. Magnetic-responsive nanoparticles are typically made of magnetic elements or compounds (such as iron, cobalt, nickel, or their alloys). By changing the time and total energy of the applied magnetic field, the magnetism of nanoparticles is controlled, thereby enhancing the conductivity of the matrix and regulating drug release in terms of time and space. In microneedle systems, magnetic nanoparticles are mainly combined with polymer materials to construct magnetic polymer composites.

Chen’s team [102] reported magnetic graphene quantum dots prepared by chemical coating of graphene with iron oxide, which have been proven to be low-toxicity at low concentrations, exhibit photoluminescence, and possess superparamagnetism, making them suitable for drug targeting and tracking. Chen et al. [102] developed a novel microneedle for transdermal drug delivery based on biodegradable chitosan and photoluminescent, superparamagnetic graphene quantum dot composite materials. The drug is bound to the graphene quantum dots in a 1:1 ratio, and then targeted release at the site is achieved through an external magnetic field. At the same time, monitoring is carried out through nuclear magnetic resonance imaging or fluorescence imaging.

Inspired by LEGO building blocks, a magnetic-responsive microneedle robot with a magnetic substrate, detachable connections, and tips has been proposed [103]. Polyethylene glycol diacrylate (PEGDA), when combined with magnetizable microparticles (NdFeB particles), can maintain high magnetization strength in an external magnetic field and is used as the material for the magnetic base. The magnetic base of the microneedle robot can be navigated and controlled through an external magnetic field. The detachable connections rapidly degrade upon contact with digestive fluids, while the drug-loaded microneedle tips penetrate the intestinal barrier and continuously release the drug. Lee et al. [104] proposed an active delivery method for multilayer drug-loaded microneedle patches using capsules that an external magnetic field can drive. Multilayer microneedle patches are attached to permanent magnets in magnetic-driven capsules. Under the influence of an external magnetic field generated by an electromagnetic drive system, the capsule with multiple layers of microneedle patches can reach the target lesion, and each microneedle patch can be delivered to the target lesion for treatment.

In summary, research on magnetically responsive polymer microneedles for transdermal drug delivery is relatively limited, with insufficient data support, and there are challenges in ensuring safety and ease of use. However, there are promising avenues for further development in this area.

As shown in Table 2, in addition to the work specifically described above, we have outlined several responsive microneedles triggered by exogenous stimuli, outlined their responsive materials, and loaded drugs and application scenarios.

## 4. Application of Smart Responsive Microneedles

### 4.1. Blood Glucose Control

Diabetes is a metabolic disorder characterized by chronically elevated blood glucose levels. As of 2021, 537 million adults worldwide were diagnosed with diabetes [113]. Improper blood sugar control can cause significant suffering for diabetes patients, leading to complications such as emaciation, glycosuria, blindness, and foot ulcers, significantly affecting their quality of life and overall health [114,115]. Due to the complex etiology of the disease, there is currently no cure for diabetes. However, research has shown that sustained and stable blood sugar control can significantly reduce the suffering of diabetes patients and prevent the onset of complications.

While some oral hypoglycemic agents (such as sulfonylureas, metformin, alpha-glucosidase inhibitors, etc.) have been used in clinical treatment, insulin is indispensable for all type 1 diabetes patients and approximately 40% of type 2 diabetes patients. Proper dosage and timing of insulin administration are crucial for regulating blood glucose levels in diabetes patients [116]. Traditional open-loop insulin delivery requires frequent blood glucose monitoring and timely subcutaneous insulin injections after meals. However, this conventional method is challenging for precisely controlling blood glucose levels, increasing the risk of diabetes complications.

Microneedles, especially smart responsive microneedles, effectively address this issue. Yu et al. [36] developed swellable glucose-responsive microneedles composed of insulin and non-degradable glucose-responsive polymer matrices to regulate blood glucose levels in diabetic mice and small pigs suffering from insulin deficiency. Under high glucose conditions, phenylboronic acid units in the polymer matrix form glucose–boronate complexes. Due to the increased negative charge, the polymer matrix expands and weakens the electrostatic interaction between negatively charged insulin and the polymer, promoting the rapid release of insulin.

To address concerns about the risk of excessive insulin release during treatment, they subsequently developed hybrid microneedle patches that simultaneously carry insulin and glucagon [21], using phenylboronic acid units as glucose-responsive elements. This microneedle patch consists of a dual module containing insulin and glucagon, mimicking the function of pancreatic cells to comprehensively regulate blood glucose levels while avoiding the risk of hypoglycemia during treatment. The prepared microneedle patch can simulate insulin or glucagon secretion by pancreatic cells in response to changes in plasma glucose levels within 6 h.

However, these passively released smart microneedles based on endogenous stimuli face challenges such as a decline in drug release rate and uneven spatial distribution. To address this issue, our team [18] developed an electrical responsive microneedle based on exogenous stimuli, using thiolated silk protein as the microneedle substrate and taking advantage of the oxidation–reduction reaction of thiol groups with disulfide bonds under electrical current to achieve reversible electrically responsive insulin release. This microneedle effectively controlled post-meal blood glucose levels by opening and closing the power supply and maintained safe blood glucose control over an extended period (11 h). This electrically responsive delivery microneedle shows potential for integration with glucose signal monitoring and establishing a closed-loop insulin delivery system.

### 4.2. Wound Repair

Chronic wound repair poses a worldwide challenge, seriously threatening patients’ lives and quality of life. As the global population ages and the number of chronic wound patients increases, this situation is expected to worsen [117]. Therapeutic molecules or growth factors for wound repair can effectively accelerate wound healing through antimicrobial action, inflammation suppression, and promoting vascular regeneration. However, the structure of the skin’s stratum corneum severely limits their delivery to damaged tissues.

Microneedles can efficiently deliver active ingredients painlessly and minimally invasively without penetrating the blood vessels, nerve fibers, or peripherals in the epidermis or dermis, thus promoting wound healing. Zhang et al. [118] loaded black phosphorus quantum dots and oxygen-carrying hemoglobin onto the tips of microneedles, achieving NIR-responsive oxygen delivery to repair diabetic wounds. These microneedles consist of PVA as the backing layer material and methylpropene gelatin (GelMA) as the tip material. After application to the skin, the PVA backing layer dissolves within minutes, leaving the GelMA tips inside the wound. When exposed to near-infrared light, black phosphorus quantum dots can rapidly convert light energy into heat, increasing local temperature, reducing hemoglobin’s oxygen-binding capacity, and leading to controlled oxygen delivery. Importantly, these microneedles demonstrated excellent wound healing capabilities when treating full-thickness skin wounds in a type 1 diabetes rat model.

In addition to being designed for transdermal drug delivery through the stratum corneum, microneedles offer several unique advantages in wound repair and tissue regeneration. Microneedles can naturally provide mechanical stimulation for wound healing and tissue regeneration. Due to their micrometer-scale needle tip array structure, microneedles can induce collagen deposition and remodeling by tissue penetration and altering the stress environment in the skin area they penetrate, reducing scarring during wound healing [119,120]. Zhang et al. [119] demonstrated that microneedle patches made from silk fibroin protein could painlessly suppress hypertrophic scarring through a minimally invasive approach. Their treatment mechanism is mainly attributed to the reduced mechanical stress and contraction produced by fibroblasts, leading to decreased expression of the mechanosensitive gene ANKRD1. Specifically, silk fibroin microneedle attenuated integrin-FAK signaling results in the downregulation of TGF-β1, α-SMA, collagen I, and fibronectin expression. This created a low-stress microenvironment, significantly reducing scar formation. Microneedles achieve scarless wound repair treatment through both smart drug delivery and physical intervention via mechanical therapy.

### 4.3. Cancer Treatment

Cancer is the second leading cause of death worldwide, with new cases increasing annually. Extracorporeal drug administration, also known as injection therapy, is currently the fastest, most direct, and most reliable method of drug delivery in cancer treatment. However, it has drawbacks such as low site-specific bioavailability, rapid drug clearance, and limited drug accumulation, reducing its effectiveness [121].

Lan et al. [122] reported a microneedle-mediated transdermal delivery of pH-responsive platinum nanoparticles for effective and safe cancer therapy, as shown in Figure 9a. They embedded pH-responsive platinum nanoparticles in carboxymethyl cellulose sodium microneedles, released after skin insertion. This allows cisplatin to be safely delivered through the stratum corneum, inhibiting cancer cell proliferation and inducing apoptosis for cancer treatment. Moreover, recent research has demonstrated that microneedle patches can be used for combination therapy in cancer and serve as a novel synergistic system, particularly in combinations of different treatments or drugs to achieve more effective therapeutic approaches [123,124]. Chen et al. [123] developed an active oxygen and NIR light-responsive microneedle system for melanoma treatment. Using PVP as the base, stable encapsulation of CuO_2_ nanoparticles in microneedles (MN@CuO_2_) catalyzes the conversion of hydrogen peroxide (H_2_O_2_) to toxic hydroxyl radicals (·OH), disrupting the redox homeostasis of tumor cells and inducing tumor cell apoptosis, as shown in Figure 9b. On the other hand, these CuO_2_ nanoparticles serve as near-infrared photothermal agents, rapidly generating significant heating in the tumor area under NIR irradiation and killing melanoma cells using photothermal therapy. This easily prepared, compositionally simple, multifunctional microneedle system holds clinical application prospects for efficient and safe melanoma treatment.

In addition to carrying small-molecule drugs and photothermal responsive elements for chemotherapy and photothermal therapy in cancer, microneedles can also carry biologically active drugs such as therapeutic immune checkpoint inhibitors (e.g., anti-CTLA-4 and anti-PD-1/PD-L1 antibodies) and CAR-T cells for cancer treatment through immunotherapy [125]. Lan et al. [126] developed a pH-responsive composite nanoparticle loaded with immune checkpoint inhibitor anti-PD-1 (aPD-1) and cisplatin, which was embedded in microneedles for transdermal drug delivery, as shown in Figure 9c. aPD-1 competitively blocks the binding of the PD-1 ligand to PD-1, leading to T cell activation. Simultaneously, the release of cisplatin intracellularly induces direct cytotoxicity to tumor cells, eradicating cancer cells and thus completing cancer treatment. Li et al. [127] constructed a tumor-responsive peptide-supramolecular composite microneedle system for melanoma treatment, as depicted in Figure 9d. Microneedles were loaded with PD-L1-targeting peptide (FE) with spherical micelle self-assembly properties. FE efficiently encapsulated adjuvants, forming a complete nanomedicine. The composite microneedles continuously released FE and decomposed into monomers, achieving PD-1/PD-L1 axis blockade while reprogramming the immunosuppressive tumor microenvironment. This synergistic therapy effectively suppressed melanoma growth, offering a new strategy for combination therapy in melanoma treatment.

**Figure 9 molecules-28-07411-f009:**
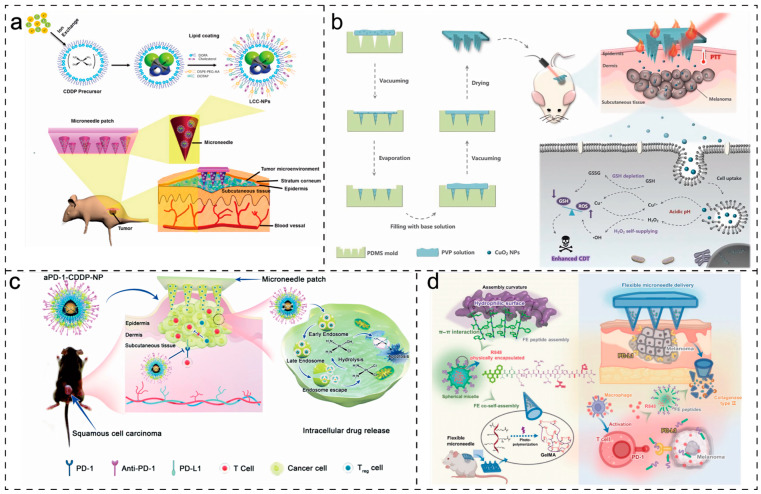
Smart microneedles for cancer treatment: (**a**) schematic representation of microneedle-mediated transdermal delivery of liposome-encapsulated cisplatin nanoparticles (lc-nps); (**b**) schematic diagram of active oxygen and NIR light-responsive microneedle system for melanoma treatment, with polyvinylpyrrolidone (PVP) as the base and stably encapsulated CuO_2_ nanoparticles; (**c**) Schematic representation of a pH-responsive microneedle loaded with aPD-1 and cisplatin composite nanoparticles; (**d**) schematic representation of a tumor-responsive peptide-supramolecular composite microneedle system for melanoma treatment. Image used with permission of [122,123,126,127].

### 4.4. Treatment of Other Diseases

Growth hormone deficiency has become a serious healthcare burden, and presents a huge impact on the physical and mental health of patients. In order to obtain a microneedle patch with long-term sustained release of growth hormone, Yang et al. [128] designed an active separation microneedle patch based on silk fibroin for the treatment of growth hormone deficiency. The backing layer of microneedles is polyacrylic acid (PAA) ethanol solution containing NaHCO_3_. When the microacupuncture is inserted into the skin, PAA absorbs the skin interstitial fluid to produce H^+^, reacts with NaHCO_3_ to produce CO_2_ and H_2_O, and realizes the active separation of MN patches. The MN patch implanted in the skin can achieve sustained release of growth hormone in rats for more than 7 days. Lee et al. [129] combined ion electroosmosis therapy with a microneedle device (Tappy Tok Tok^®^, U-BioMed Inc., Daegu, Republic of Korea) in order to increase the delivery of recombinant human growth hormone, and showed that anodic ion electroosmosis therapy with a higher current density enhanced the penetration of growth hormone through microneedle-generated microchannels.

Duong et al. [130] used DNA vaccines and loaded on microneedles to achieve pH response release. To control the release of the DNA vaccine, polycarbonate micronicles were coated with a positively charged super-pH responsive oligo-amino-ethylamine-coupled polymer (β-amino-carbamate) (OSM-(PEG-PAEU)) and a negatively charged immunostimulating adjuvant poly(I:C) under low pH conditions. After implantation into the skin, due to charge reversal and electrostatic rejection, the DNA vaccine is released, which enhances dendritic cell maturation and induces type I interferon, thus producing antibodies. In addition, microneedles store vaccines in a solid dry form, which increases the thermal stability of vaccines and facilitates mass vaccination of vaccines in developing countries that lack effective cold-chain transportation and storage, and lack trained health care professionals [131,132]. On top of that, the skin is full of immune cells targeted by immunotherapy, which produces a stronger immune effect than a subcutaneous injection.

In addition to being used in biomedicine, intelligent responsive microneedles also have applications in aesthetic medicine. Fang et al. [133] used 3D printing technology to manufacture a magnetically responsive transdermal composite microneedle with mesoporous iron oxide nanoraspberry (MIO) loaded with minoxidil. Embedded in the tip of the microneedles, MIO can deliver minoxidil and generate mild heat to promote hair growth. Ten days after receiving microneedle treatment, the mice showed an 800% improvement in hair density compared to mice that received no treatment.

## 5. Development Trends and Clinical Transformation

In recent years, microneedles have gained novel and intriguing features such as rapid dissolution/swelling, separability, competent responsiveness, and antimicrobial properties to meet the complex demands in practical applications. Stimulus-responsive microneedle patches are an emerging technology for on-demand drug delivery to enhance treatment outcomes and maximize patient compliance. Encouraging results and significant potential have been observed for smart responsive microneedles in the field of drug delivery.

From a regulatory perspective, most of the current applications of microneedles require rigorous clinical trials, and non-phased trials ensure the physical and physiological safety and effectiveness of responsive microneedles. At the same time, most microneedles are loaded with drugs, so the stability and activity of drugs in the process of production and storage require strict production specification auditing. According to the classification of medical devices by the Food and Drug Administration, microneedle drug patch tablets (without drug loading) are regarded as Class II medical devices. Clinical trials of the recently launched stimulus-responsive microneedles are expected to start in the near future.

Continued research efforts are required in the following directions to accelerate the adoption of smart responsive microneedles into the medical market. The first issue concerns the choice of microneedle materials. Many researchers have employed some relatively novel stimulus-responsive polymers to prepare smart responsive microneedles to achieve more efficient and durable smart responsive effects. While these innovative synthetic materials show promise, their safety may not be thoroughly proven, as they are often not approved pharmaceutical excipients. This poses a potential regulatory hurdle for clinical translation, and pharmaceutical companies may need to invest considerable time and resources to obtain regulatory approvals for these new excipients. In addition to meeting the mechanical performance and safety of microneedles, the materials for microneedles must also ensure the activity of the loaded drugs during preparation, storage, and delivery, a factor often overlooked by many researchers. This becomes particularly critical as many innovative microneedle studies attempt to deliver biologically active macromolecules, highlighting the importance of developing materials to stabilize the drugs.

The second issue is the rate and effectiveness of microneedle responsiveness. Scientists still need to gain a deeper understanding of the pathological environment within the human body to develop more effective smart responsive strategies that meet the personalized needs of patients. One potential direction is the development of multi-responsive smart microneedle systems that combine multiple responsive strategies to enable smart delivery in response to different stimuli or multiple drugs, thereby expanding their applicability to a broader range of diseases or preventive measures. Another potential direction is the integration of microneedles for monitoring purposes, developing smart microneedles with dual capabilities for drug delivery and physiological monitoring to achieve precise drug delivery.

The third issue remains safety concerns. Standardization of production and regulation of smart responsive microneedles is necessary. While most studies related to smart responsive microneedles have explored their biocompatibility to some extent, foundational research is still required. This includes comprehensive investigations into biocompatibility factors such as cytotoxicity, acute/chronic systemic toxicity, hemolysis tests, irritation tests, implantation tests, intradermal reactivity tests, biodegradation tests, and carcinogenicity tests. Ensuring the safety of using smart responsive microneedles still necessitates thorough validation in different animal or preclinical models.

## 6. Conclusions

This article provides a comprehensive overview of the critical research advancements in smart responsive microneedles for drug delivery since their inception. It details various types and mechanisms of smart responsive microneedles’ endogenous and exogenous stimulus-responsive behaviors and summarizes their current applications. Notably, the article also looks ahead to the future directions of smart responsive microneedles for drug delivery. We believe that with continued progress in effectiveness and safety, intelligent responsive microneedles will soon find their systematic avenue into medical applications, benefiting a larger population of patients.

## Figures and Tables

**Figure 1 molecules-28-07411-f001:**
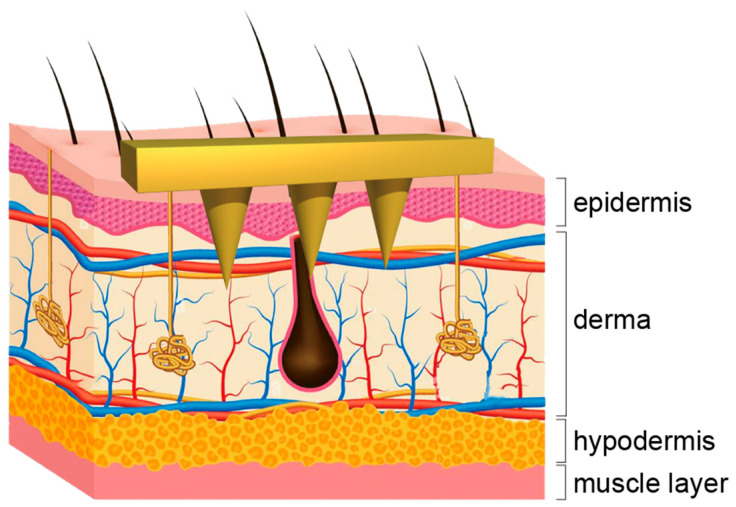
Schematic diagram of microneedles for transdermal drug delivery.

**Figure 2 molecules-28-07411-f002:**
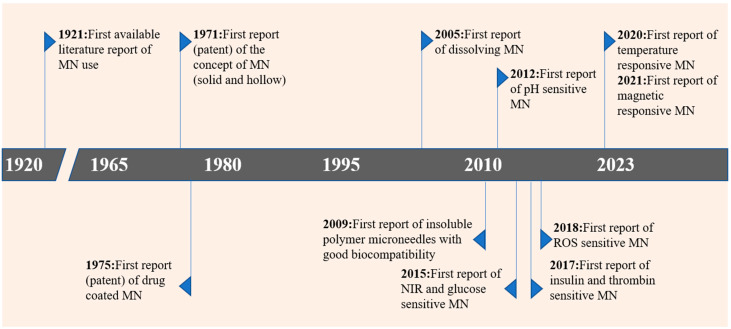
Timeline of microneedle development.

**Figure 3 molecules-28-07411-f003:**
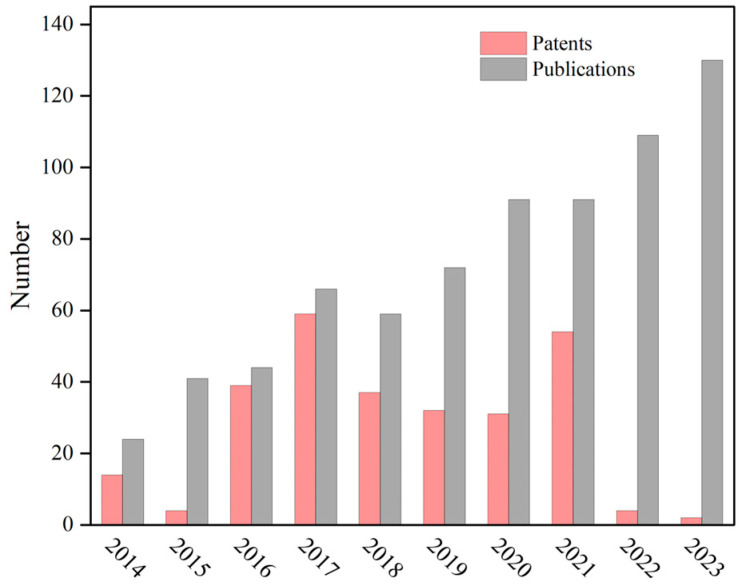
Number of publications and patents based on responsive microneedles in the last 10 years.

**Figure 4 molecules-28-07411-f004:**
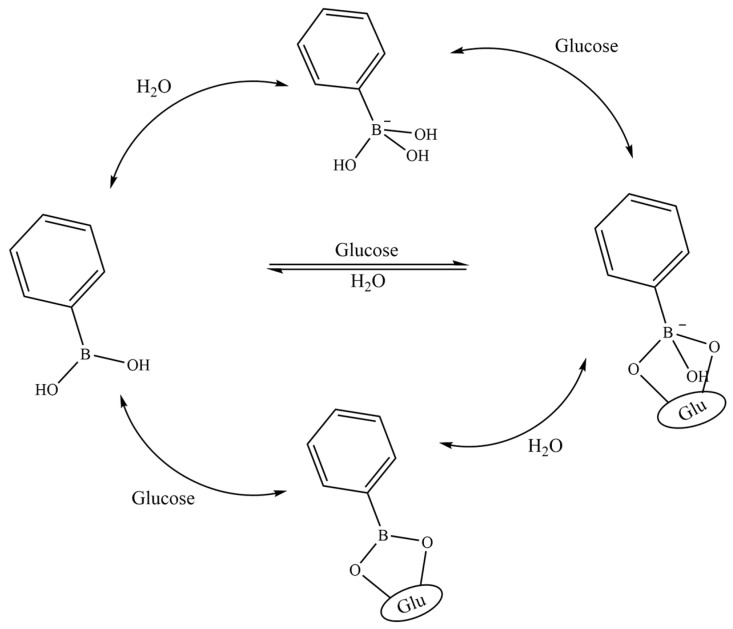
Schematic diagram of specific binding of phenylboric acid and glucose. Image used with permission of [35].

**Figure 5 molecules-28-07411-f005:**
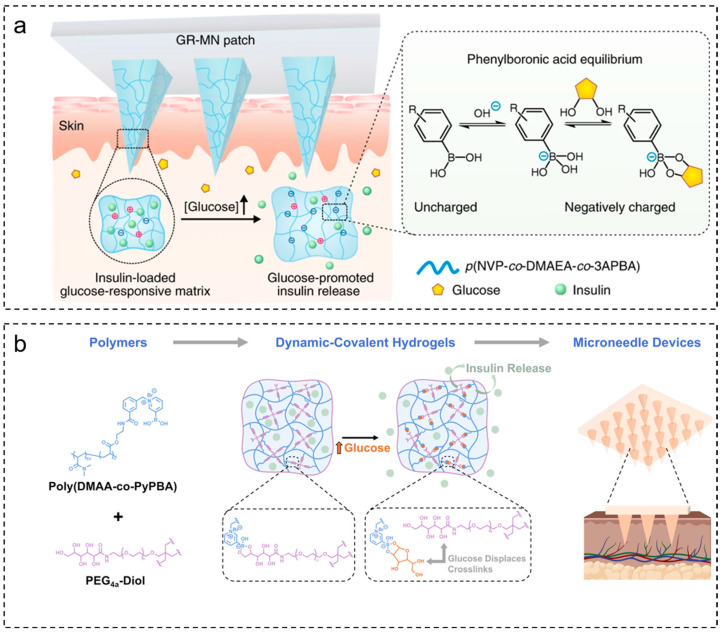
Preparation of glucose-sensitive microneedles based on PBA. (**a**) When the glucose concentration increases, the increase in negative charge of the glucose-PBA complex weakens the electrostatic interaction between insulin and the polymer, thus promoting the release of insulin. (**b**) A synthetic polymer with a PBA motif is combined with four-arm polyethylene glycol to form a dynamic covalent PBA-diol crosslinked hydrogel determined by glucose levels. After the preparation of microneedles, insulin release in microneedles is accelerated in the presence of glucose. Images used with permission of [36,37].

**Figure 6 molecules-28-07411-f006:**
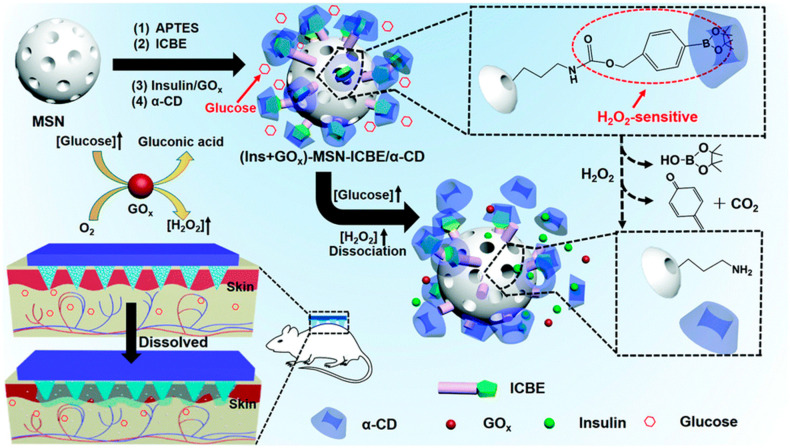
Schematic diagram of mesoporous silica nanoparticles integrated with a microneedle patch for glucose-responsive transdermal insulin delivery. Image used with permission of [46].

**Figure 8 molecules-28-07411-f008:**
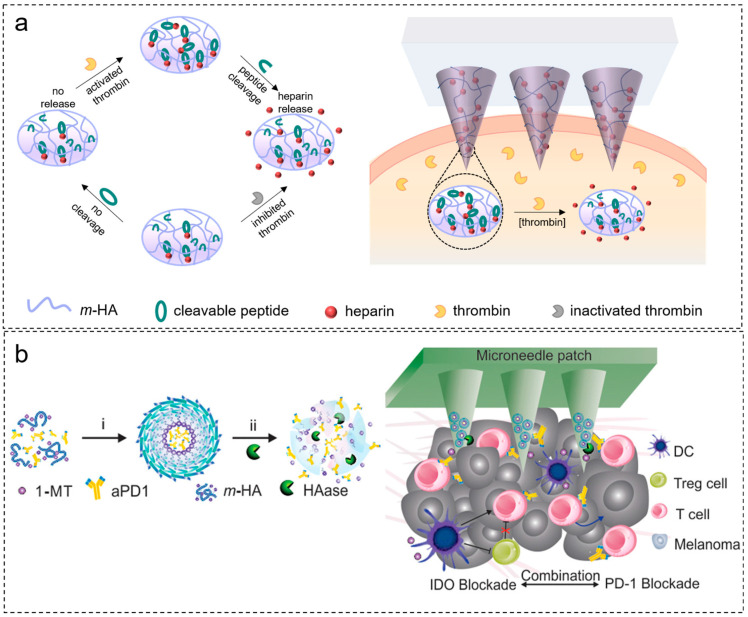
Preparation of enzyme-responsive microneedles: (**a**) preparation of thrombin-responsive microneedles for continuous regulation of blood clotting; (**b**) preparation of hyaluronidase-responsive microneedle patches for enhancing antitumor and T-cell immunity. Image used with permission of [70,71].

**Table 1 molecules-28-07411-t001:** Responsive microneedles triggered by endogenous stimuli.

Responsive Materials	Stimulus Types	Loaded Drugs	Applications	References
3-carboxy-4-fluorophenylboronicacid	Glucose	Porcine insulin	Blood glucose control	[73]
3-aminophenylboronic acid	Glucose	Bovine insulin	Blood glucose control	[74]
4-(2-acrylamidoethylcarbamoyl)-3-fluorophenylboronic acid	Glucose	Gluconic insulin	Diabetic wound	[75]
GOx/poly(ethylene glycol)-*b*-PHMEMA	Glucose/pH	Insulin	Blood glucose control	[22]
3-carboxy-4-fluorophenylboronicacid	Glucose	Porcine insulin	Blood glucose control	[76]
2,3-dimethylmaleic anhydride-polyethyleneimine-polylactic acid-glycolic acid copolymer	pH	Antimicrobial peptide	Chronic wound healing	[77]
Cellulose acetate phthalate	pH	N/A	N/A	[78]
NaHCO_3_	pH	Alexa 488/Cy5	N/A	[79]
Pyridine	pH	ovalbumin	N/A	[80]
Polydopamine	pH	DNA vaccine	Vaccine delivery	[81]
Oligo(sulfamethazine)-b-poly(ethylene glycol)-b-poly(amino urethane)	pH	DNA vaccine	Vaccine delivery	[82]
Copolymer methyl ether poly(ethylene glycol)-poly(β-aminoester)	pH	AIEgen (NIR950)	Tumor therapy	[83]
Thioketal	ROS	Methotrexate	Psoriasis therapy	[84]
Boronate moiety	ROS	Doxorubicin	Melanoma therapy	[85]
N1-(4-boronobenzyl)-N3-(4-boronophenyl)-N1, N1, N3, N3-tetramethylpropane-1,3-diaminium	ROS	avβ6-blocking antibody [10D5]	Pulmonary fibrosis	[86]
1-methyl-DL-tryptophan-conjugated hyaluronic acid	Hyaluronidase	Anti-PD1 antibody	Cancer immunotherapy	[87]
Hyaluronic acid methacrylate	Hyaluronidase	Basic fibroblast growth factor	Bacterial infection/diabetic wounds	[88]

**Table 2 molecules-28-07411-t002:** Responsive microneedles triggered by exogenous stimuli.

Responsive Materials	Stimulus Types	Loaded Drugs	Applications	References
Polycaprolactone	Temperature	Metformin	Diabetes	[105]
Poloxamer formulations	Temperature	Fluorescein sodium	sustained ocular drug delivery	[106]
MXene	NIR	Mupirocin/human epidermal growth factor	Wound management	[107]
Graphene oxide	NIR	Nitric oxide	Wound healing	[108]
Ag/AgCl electrodes	Electric	Ovalbumin	COVID-19	[109]
Electrode array	Electric	siRNA(PD-L1) alone or combined with aPD-1 or immunoadjuvant of CPG2395	Cancer immunotherapy	[110]
Polycaprolactone/hyaluronic acid	Machine	Vaccine antigen of canine influenza virus	Vaccine immunity	[111]
Asymmetric microneedles loaded with *Schistosoma japonicum* egg tips on CA-CMC	Machine	*Schistosoma japonicum* egg	Type 1 diabetes mellitus	[112]

## Data Availability

Not applicable.

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
