# Peer review of "Smart Responsive Microneedles for Controlled Drug Delivery"

_molecules, 2023, doi:10.3390/molecules28217411_

Round 1

Reviewer 1 Report

Comments and Suggestions for Authors

Manuscript is well drafted by author and could be the useful literature for the readers working in the area of smart responsive microneedle technology. I am suggesting some minor revision

1. Fabrication and characterization of microneedles should be included in manuscript with reference to following articles

https://doi.org/10.3390/pharmaceutics14051097

https://doi.org/10.3390/scipharm91020027

2. Write about regulatory aspects about these smart responsive microneedles along with patent scenario

3. Include some global statistics related to biomedical use of these smart responsive microneedles

4. Minor english grammar editing is required

Comments on the Quality of English Language

Minor english grammar editing is required

Author Response

Comments:

Manuscript is well drafted by author and could be the useful literature for the readers working in the area of smart responsive microneedle technology. I am suggesting some minor revision.

We appreciate the reviewer’s positive evaluation of our work. Thanks very much for taking your time to review this manuscript. We really appreciate all your generous comments and suggestions! Please find my revisions in the re-submitted files.

  1. Fabrication and characterization of microneedles should be included in manuscript with reference to following articles.

https://doi.org/10.3390/pharmaceutics14051097

https://doi.org/10.3390/scipharm91020027

Thank you for your advice. We have added a brief Introduction to The preparation of microneedles in Introduction“The earliest preparation method of microneedles is MEMS technology, and on this basis, laser cutting, photolithography, wet and dry etching, micro-molding, 3D printing and so on have been developed [14, 15]. One of the most common, simplest and most conven-ient methods is the micro-molding method.” Thank you very much for the literature on the preparation and characterization of microneedles provided by you. I believe that the supplement of these contents will make this manuscript more comprehensive. At the same time, we also pointed out in Development Trends and Clinical Transformation that the quality supervision of microneedles should include its physical properties, stability, biosafety and effectiveness. It is worth noting that the preparation and characterization of responsive microneedles does not seem to be much different from that of traditional microneedles. Due to limited space, this manuscript focuses on the introduction of the special properties of responsive microneedles, and does not focus on the preparation and characterization of microneedles.

  1. Write about regulatory aspects about these smart responsive microneedles along with patent scenario

In Development Trends and Clinical Transformation, we mentioned that microneedles regulation must ensure their physical and physiological safety and efficacy, as well as the activity of the drugs on which microneedles are loaded.

Through the reference about microneedle patent (DOI: 10.1016 / j.biomaterials. 2020.120491, DOI:10.1080/13543776.2020.1742324), using the Cooperative Patent Classification System (CPC System) developed by the European Patent Office (EPO) and the United States Patent and Trademark Office (USPTO) and the database of the World Intellectual Property Organization, We used the keywords "microneedle" and "smart"/" responsive "to conduct a preliminary search on the patent applications of intelligent responsive microneedle in the past 10 years, and summarized the search situation, and obtained the following data.

There may be some duplication, as patents and intellectual property rights in different countries and regions are not always interchangeable. We can see that microneedle patent applications reached a peak in 2017-2021, with fewer patents in recent years, of course, perhaps because the latest data is not updated.

The number of publications and patents on responsive microneedles have been supplemented in Introduction.

The number of publications and patents on responsive microneedles has increased significantly in recent years, and we obtained the following data by conducting keyword searches on Pubmed and CPC systems (Figure 3). Number of patent applications has been relatively low in recent years, perhaps because the latest data have not been up-dated and patents take longer to review.

Figure 3. Number of publications and patents based on responsive microneedles in the last 10 years.

  1. Include some global statistics related to biomedical use of these smart responsive microneedles

Latest reports of microneedle delivery therapy in the clinical application of referring to the 127 clinical trials regarding microneedles (DOI: 10.1016 / j. j conrel. 2023.07.023). A search of the database referenced (clinicaltrials.gov) did not find any clinical trials that have been conducted or are ongoing for smart response microneedles. This can be attributed to the fact that most of the current studies on intelligent response microneedles are preclinical studies, just as drug-loaded coated/hollow/solid microneedles appeared in the 1970s, but relevant clinical trials did not begin until the 2000s. One of the purposes of this review is to summarize the existing preclinical studies of intelligent response microneedle therapy, and propose the current development trend and the direction of efforts to be made in order to start clinical trials more quickly.

  1. Minor english grammar editing is required

 We checked the whole article again and made the relevant syntax corrections.

Reviewer 2 Report

Comments and Suggestions for Authors

Smart Responsive microneedles for controlled drug delivery

The current review by Shenzhou Lu et al., reported a comprehensive review in responsive microneedle systems for controlled drug delivery. The authors have summarized different responsive microneedles to various internal and external stimuli. Further, the authors have discussed the substrate, different fabrication methods, different mechanisms and their possible applications in glucose sensing, wound repair and cancer. Microneedle technology is currently an trending transdermal delivery technology per drug and vaccine delivery as it is a needleless technology that can deliver vaccine or drugs in a non-medical set-up.

Lu Lab has done substantial work in microneedle patches and their application in drug delivery and hence are well placed to write this review. The review is very well written; However, I have few minor concerns before I accept it for publication.

1)      Abstract: A clear abstract detailing the focus of the review is required.

2)      This comprehensive review needs a perspective and outlook section before the conclusion and possible future trends for this technology. It would be great to know the author’s perspective on the future of this transdermal technology.

3)      The authors have clearly covered most of the literature. However, some important reference for microneedle based delivery (drug, vaccine, gene) are missing. The authors should include these references and mention them in the discussion within the review.

1)Silk Fibroin Microneedle Patches for the Sustained Release of Levonorgestrel

ACS Appl. Bio Mater. 2020, 3, 8, 5375–5382

2) Silk Fibroin Microneedles for Transdermal Vaccine Delivery

ACS Biomater. Sci. Eng. 2017, 3, 3, 360–369

3) Fabrication of Silk Microneedles for Controlled-Release Drug Delivery

Advanced Functional Material 2012

4) Transdermal microneedles for the programmable burst release of multiple vaccine payloads

Nature Biomedical Engineering volume 5, pages998–1007 (2021)

5) Microneedle-mediated gene delivery for the treatment of ischemic myocardial disease

SCIENCE ADVANCES, 2020 Vol 6, Issue 25DOI: 10.1126/sciadv.aaz3621

Comments on the Quality of English Language

Reviewer 3 Report

Comments and Suggestions for Authors

In this work, the author reviewed recent development in smart responsive polymer microneedles at home and abroad. It summarizes the response mechanisms based on various stimuli and their respective application scenarios. It's timely and well-organized. However, there are several minor issues should be addressed before acceptance.

1. Microneedles are not only used for blood glucose control,  wound repair and cancer treatment, but also in other fields. The author should discuss in detail the application of microneedles in other areas such as gout management, and growth hormone deficiency, etc.  The author can refer to and cite the following references (Yang et al. Acta Pharm Sin B. 2023,13(1):344-358; Yang et al. Asian J Pharm Sci. 2022,17(1):70-86; Yang et al. Acta Pharm Sin B. 2023,13(8):3454-3470). 

2. The quality of images in Figure 6 and Figure 8 should be improved. Some images can not be seen clearly.

3. It's better that the author use a schematic diagram to represent various typical endogenous and exogenous responsive smart microneedles.
